# Dual-light photodynamic therapy administered daily provides a sustained antibacterial effect on biofilm and prevents *Streptococcus mutans* adaptation

**Sakari Nikinmaa**[1,2], **Heikki Alapulli**[3], **Petri Auvinen**[4], **Martti Vaara**[5,6], **Juha Rantala**[2], **Esko Kankuri**[7], **Timo Sorsa**[3,8], **Jukka Meurman**[1,3], **Tommi Pätilä**[1,2,9]*

**1** Department of Neuroscience and Biomedical Engineering, Aalto University, Espoo, Finland, **2** Koite Health Oy, Espoo, Finland, **3** Department of Oral and Maxillofacial Diseases, University of Helsinki, Helsinki, Finland, **4** Institute of Biotechnology, University of Helsinki, Helsinki, Finland, **5** Northern Antibiotics, Espoo, Finland, **6** Department of Bacteriology and Immunology, University of Helsinki, Medical School, Helsinki, Finland, **7** Department of Pharmacology, University of Helsinki, Helsinki, Finland, **8** Department of Oral Diseases, Karolinska Institute, Huddinge, Sweden, **9** Department of Congenital Heart Surgery and Organ Transplantation, New Children's Hospital, University of Helsinki, Helsinki, Finland

* tommi.patila@hus.fi

**Data Availability Statement:** All relevant data are within the paper.

## Abstract

Antibacterial photodynamic therapy (aPDT) and antibacterial blue light (aBL) are emerging treatment methods auxiliary to mechanical debridement for periodontitis. APDT provided with near-infrared (NIR) light in conjunction with an indocyanine green (ICG) photosensitizer has shown efficacy in several dental in-office-treatment protocols. In this study, we tested *Streptococcus mutans* biofilm sensitivity to either aPDT, aBL or their combination dual-light aPDT (simultaneous aPDT and aBL) exposure. Biofilm was cultured by pipetting diluted *Streptococcus mutans* suspension with growth medium on the bottom of well plates. Either aPDT (810 nm) or aBL (405 nm) or a dual-light aPDT (simultaneous 810 nm aPDT and 405 nm aBL) was applied with an ICG photosensitizer in cases of aPDT or dual-light, while keeping the total given radiant exposure constant at 100 J/cm$^2$. Single-dose light exposures were given after one-day or four-day biofilm incubations. Also, a model of daily treatment was provided by repeating the same light dose daily on four-day and fourteen-day biofilm incubations. Finally, the antibacterial action of the dual-light aPDT with different energy ratios of 810 nm and 405 nm of light were examined on the single-day and four-day biofilm protocols. At the end of each experiment the bacterial viability was assessed by colony-forming unit method. Separate samples were prepared for confocal 3D biofilm imaging. On a one-day biofilm, the dual-light aPDT was significantly more efficient than aBL or aPDT, although all modalities were bactericidal. On a four-day biofilm, a single exposure of aPDT or dual-light aPDT was more efficient than aBL, resulting in a four logarithmic scale reduction in bacterial counts. Surprisingly, when the same amount of aPDT was repeated daily on a four-day or a fourteen-day biofilm, bacterial viability improved significantly. A similar improvement in bacterial viability was observed after repetitive aBL application. This viability improvement was eliminated when dual-light aPDT was applied. By changing the 405 nm to

**Funding:** This paper is part of SN PhD studies at the Aalto University, Espoo, Finland. Koite Health has provided support in the form of salaries for author [SN] but did not have any additional role in the study design, data collection and analysis, decision to publish, or preparation of the manuscript. The specific roles of these authors are articulated in the 'author contributions' section.

**Competing interests:** We have a financial disclosure about the paper, including authors, Sakari Nikinmaa, Tommi Pätilä and Juha Rantala. These authors are shareholders in a company Koite Health Oy, where SN and TP are also members of the board. Koite Health has filed patents P21233F100 and P22769F100, which are related to antibacterial dual light. The company Koite Health is developing a dual light antibacterial product for prevention of dental infections. Martti Vaara is a shareholder and a board member in Northern Antibiotics INC, which is dedicated to developing novel Colistin antibiotics. This financial disclosure does not alter our adherence to PLOS ONE policies on sharing data and materials.

810 nm radiant exposure ratio in dual-light aPDT, the increase in aBL improved the antibacterial action when the biofilm was older. In conclusion, when aPDT is administered repeatedly to *S. mutans* biofilm, a single wavelength-based aBL or aPDT leads to a significant biofilm adaptation and increased *S. mutans* viability. The combined use of aBL light in synchrony with aPDT arrests the adaptation and provides significantly improved and sustained antibacterial efficacy.

## Introduction

Oral hygiene based on mechanical cleansing by removal of the biofilm has been proven to be the best method for the prevention of dental and periodontal disease [1]. While saliva contains some 700 different bacterial species, regularly performed mechanical biofilm removal essentially only leaves early-forming Streptococcal residual biofilm on the dental surface [2]. Dental and periodontal diseases result from prolonged biofilm infections [3]. Neglected hygiene is usually a factor in the complex multispecies biofilm required for the disease process. While caries essentially involves mainly gram-positive, *Streptococcus*-rich and carbohydrate fermenting biofilm, gingivitis, and periodontitis are related to gram-negative, proteolytic bacterial biofilm flora [4]. Sometimes genetic or environmental circumstances, such as virulent bacterial strains, may predispose one to disease despite reasonable dental hygiene [5].

Antibacterial photodynamic therapy (aPDT) and antibacterial blue light (aBL) have emerged as solutions for attacking dental biofilm [6,7]. These methods are based on light photon absorption by chromophores, leading to electron transfer reactions that ultimately result in the production of reactive oxygen species (ROS). Similarly, ROS are used in bacterial killing by polymorphonuclear leucocytes in the phagocytotic bacterial elimination process [8]. Antibacterial PDT combines an externally provided photo enhancer with a specific light wavelength to excite the nearby oxygen into a singlet state, which is mostly responsible for the antibacterial effect. Antibacterial blue light, however, is based on the same mechanism, but the electron transfer reaction occurs by inherent photosensitizers found within the bacteria themselves, mostly porphyrins and flavins. Certain *Streptococcus* species, including *S. mutans*, are vulnerable to aPDT due to poor ROS scavenging capacity, mostly reflecting the lack of catalase enzyme. On the other hand, aBL is particularly effective for those bacteria carrying the most abundant amount of blue-light absorbing porphyrins, such as the cell-surface black pigment, iron protoporphyrin IX, in the so-called black-pigmented bacteria group [9,10].

Indocyanine green (ICG) is a widely used aPDT photosensitizer in dentistry due to its low toxicity, non-ionizing properties, water solubility and light absorption at near-infrared (NIR) wavelengths, which have a good tissue penetration. Several studies have shown the efficacy of NIR 810 nm/ICG aPDT as an adjunctive periodontal treatment. In these studies, dosing has been infrequent, mostly due to the light administration requiring a dentist's in-office equipment and expertise. Auspiciously, rapid development in light-emitting diode (LED) technology has allowed for the development of personal products for a light application used at home. In a home setting, the aPDT treatment can be self-administered by the patients themselves, on a more regular and frequent basis.

The antimicrobial efficacy of aPDT is evident for planktonic bacteria, whereas in biofilms, bacteria are more resistant to any antibacterial treatment [11–13]. Furthermore, data on the effect of repetitive aPDT is sparse. Generally, bacteria are unable to produce resistance against aPDT and aBL, although some adaptation has been shown [7,14–17]. In this study, we tested

the efficacy and effect sustainability of aPDT and aBL in an *S. mutans* biofilm-model allowing for repeated daily treatment administrations. We compared aPDT or aBL head-to-head, and assessed the effect of their simultaneous application, called dual-light aPDT treatment. We also examined the bactericidal effect of different ratios of aPDT and aBL when the dual-light aPDT was applied. Finally, to analyze the aPDT action mechanism, we evaluated ICG adhesion to *S. mutans* bacteria, using absorption spectroscopy.

## Materials and methods

Monospecies *S. mutans* biofilm model experiments were performed to study the effect of recurring photodynamic therapy during the biofilm formation process. A minimum of six biofilms for each experiment were grown in surface-treated, flat-bottom Nunclon Delta well plates (Thermo Fisher Scientific Inc, US), which have widely been used for *in vitro* biofilm formation of multiple different bacteria species [10,13,18], specifically, the *S. mutans* species [18–20]. All the replicants in each study protocol were performed during the same day, simultaneously. This enabled to use the same *S. mutans* suspension, identical light exposure parameters, and provided exactly similar laboratory conditions. The biofilm experiments were divided into different setups based on biofilm maturation age and the therapy given.

### Study protocols

Applications of both single- and dual-light therapies were scheduled to mimic the daily use of antibacterial light therapy at home for hygiene purposes. In all the experiments, the dosing was kept the same, meaning the total amount of light irradiance, the respective light energy applied, and the concentration of the ICG photosensitizer (if given) were identical. In the first setting, *S. mutans* biofilm was incubated for either one day or four days, and a single dose of light with ICG was applied at the end of the growth period. In the second setting, the biofilm was incubated for either four days or fourteen days, and a daily dose of light with ICG was applied during incubation. In each setting, the last treatment was followed by plating each well onto separate brain heart infusion (BHI)-agar dishes for colony-forming unit (CFU) counting. Study protocols are presented in Table 1.

### Biofilm model

*Streptococcus mutans* (ATCC 25175) bacteria were grown for 18 h in an incubator (NuAire DH autoflow 5500, NuAire inc, US), at +36 degrees C, 5% $CO_2$ in BHI broth (Bio-Rad 3564014, Bio-Rad Laboratories, Inc, US). The resulting bacterial suspension was diluted with a 0.9% NaCl solution until an optical density (OD) of 0.46 was reached. The optical density was measured by a spectrophotometer (Varian Cary 100 Bio UV-VIS, Agilent Technologies, Inc, US), and then with a Den 1 McFarland Densitometer (Biosan, Riga, Latvia).

Biofilms were grown in flat-bottom 96-well plates (Thermo Fisher Scientific Inc, US) by placing 100 μl of 0.46 OD *S. mutans* suspension in each well, with 100 μl of BHI-broth growth medium. The well plates were then incubated in a growth chamber (36°C, 5% $CO^2$). The BHI-broth medium was changed daily to supply fresh growth medium and to wash away the debris. The change of the medium in each well was performed by removing 100 μl of the medium and replacing it with a similar amount of fresh BHI broth.

### Light exposure

zBefore the light exposure took place, the growth medium was meticulously removed by pipetting and subsequently replaced with an equal amount of indocyanine green solution (Verdye,

**Table 1. Study protocols.**

| Experiment | Figure | Repeats | Number of treatments | Radiant exposure (J/cm^2) | Wavelenghts (nm) | Irradiance 405 nm (mW/cm^2) | Irradiance 810 nm (mW/cm^2) | ICG (+/-) | Biofilm age at the end of experiment (d) |
|---|---|---|---|---|---|---|---|---|---|
| aBL 1d | 1 | 6 | 1 | 100 | 405 | 80 | 0 | - | 1 |
| aPDT 1d | 1 | 6 | 1 | 100 | 810 | 0 | 100 | + | 1 |
| Dual-light 1d | 1 | 6 | 1 | 100 | 405+810 | 50 | 50 | + | 1 |
| Control 1d | 1 | 5 | N/A | N/A | N/A | N/A | N/A | - | 1 |
| 4 d single-dose aBL | 2 | 6 | 1 | 100 | 405 | 80 | 0 | - | 4 |
| 4 d single-dose aPDT | 2 | 6 | 1 | 100 | 810 | 0 | 100 | + | 4 |
| 4 d dual-light single dose | 2 | 6 | 1 | 100 | 405+810 | 50 | 50 | + | 4 |
| 4 d daily-dose aBL | 2 | 6 | 4 | 100 | 405 | 80 | 0 | - | 4 |
| 4 d daily-dose aPDT | 2 | 6 | 4 | 100 | 810 | 0 | 100 | + | 4 |
| 4 d dual-light daily dose | 2 | 6 | 4 | 100 | 405+810 | 50 | 50 | + | 4 |
| 4 d control | 2 | 12 | N/A | N/A | N/A | N/A | N/A | - | 4 |
| 14 d daily-dose aBL | 3 | 6 | 14 | 100 | 405 | 80 | 0 | - | 14 |
| 14 d daily-dose aPDT | 3 | 6 | 14 | 100 | 810 | 0 | 100 | + | 14 |
| 14 d dual-light daily 2dose | 3 | 6 | 14 | 100 | 405+810 | 50 | 50 | + | 14 |
| 14 d control | 3 | 6 | N/A | N/A | N/A | N/A | N/A | - | 14 |
| 1 d single dose 1:3 | 4 | 12 | 1 | 100 | 405+810 | 42 | 135 | + | 1 |
| 1 d single dose 1:1 | 4 | 12 | 1 | 100 | 405+810 | 73 | 79 | + | 1 |
| 1 d single dose 3:1 | 4 | 12 | 1 | 100 | 405+810 | 130 | 38 | + | 1 |
| Control 1d | 4 | 6 | N/A | N/A | N/A | N/A | N/A | - | 1 |
| 4 d single dose 1:3 | 4 | 12 | 1 | 100 | 405+810 | 42 | 135 | + | 4 |
| 4 d single dose 1:1 | 4 | 12 | 1 | 100 | 405+810 | 73 | 79 | + | 4 |
| 4 d single dose 3:1 | 4 | 12 | 1 | 100 | 405+810 | 130 | 38 | + | 4 |
| 4 d daily dose 1:3 | 4 | 6 | 4 | 100 | 405+810 | 42 | 135 | + | 4 |
| 4 d daily dose 1:1 | 4 | 6 | 4 | 100 | 405+810 | 73 | 79 | + | 4 |
| 4 d daily dose 3:1 | 4 | 6 | 4 | 100 | 405+810 | 130 | 38 | + | 4 |
| 4 d control | 4 | 3 | N/A | N/A | N/A | N/A | N/A | - | 4 |
| aPDT | 7 | 5 | 1 | 100 | 810 | 0 | 100 | + | 0 |
| Control | 7 | 3 | 1 | 100 | 810 | 0 | 100 | - | 0 |

Diagnostic Green, GmBH), tittered to a concentration of 250 μg/ml. Absorption spectrum of ICG is provided below. The indocyanine green was left to incubate at room temperature and in the dark for 10 minutes. After this incubation period, the biofilm was washed with a 0.9% NaCl solution. Then, the 0.9% solution of NaCl was added to each well to reach a total volume of 200 μl. Light exposure was performed using specific, custom-made LED light sources (Lumichip Oy, Espoo, Finland). The exposure time was calculated from the determined light amount and known irradiances, which had been previously measured with a light energy meter (Thorlabs PM 100D with S121C sensor head, Thorlabs Inc, US) and a spectroradiometer (BTS256, Giga-hertz-Optik GmbH, Germany), respectively. After the exposure, the BHI broth was changed. The emission spectra of the used light sources are presented in Table 2. The plates were then

**Table 2. The emission spectra of the used light sources.**

| Wavelength (nm) | Peak Wavelength (nm) | 50% output low side (nm) | 50% output high side (nm) |
|---|---|---|---|
| 405 | 404 | 397 | 412 |
| 810 | 811 | 793 | 825 |

placed in the incubator, or, if the light exposure was final, the biofilm was removed for CFU counting, as described below. Excitation lights were applied with two single-wave LED light sources with peak intensities at 810 nm or at 405 nm, and with a dual-wave LED light chip simultaneously producing two separate peak intensities at 405 nm and at 810 nm.

Antibacterial photodynamic therapy light exposure was administered at an 810-nm peak wavelength LED array on top of the well plate. The resulting light irradiance was 100 mW/cm$^2$, and the provided light energy was 100 J/cm$^2$. Antibacterial blue light was administered at a 405 nm peak wavelength LED array, with a resulting irradiance of 80 mW/cm$^2$, and resulting light energy of 100 J/cm$^2$. The dual light was administered with two light peaks identically placed and providing LED arrays on top of the well plate, producing a synchronous irradiance of 50 mW/cm$^2$ for the 405 nm light, and 50mW/cm$^2$ for the 810-nm light. The light energies produced were 50J/cm$^2$ (405 nm) and 50J/cm$^2$ (810 nm), respectively. To rule out the sample heating and subsequent effect on bacterial viability, temperature controls were measured (Omega HH41 Digital Thermometer, Omega Engineering, US) in the biofilm wells to confirm temperature levels below 35 degrees during the treatment, with a 100 mW/cm$^2$ radiant flux.

We also tested the antibacterial efficacy of dual-light treatment in terms of different radiant exposure ratios of 405 nm to 810 nm, when the total amount of light was kept constant at 100 J/cm$^2$. Three different light combinations were employed, with simultaneous use of the single-peak-emitting light sources. Firstly, a 1:1 radiant exposure ratio of aBL to aPDT was applied, with 70 mW/cm$^2$ irradiance for the 405 nm light and 70 mW/cm$^2$ for the 810 nm light, radiant exposures being at 50J/cm$^2$ and at 50J/cm$^2$, respectively. Secondly, a 3:1 radiant exposure ratio of aBL to aPDT was applied, with 130 mW/cm$^2$ irradiance for the 405nm light and 40mW/cm$^2$ for the 810nm light, the radiant exposures provided being at 75J/cm$^2$ and at 25J/cm$^2$, respectively. Thirdly, a 1:3 radiant exposure ratio of aBL to aPDT was applied, with 40mW/cm$^2$ irradiance for the 405 nm light and 130 mW/cm$^2$ for the 810 nm light, the radiant exposures being at 25 J/cm$^2$ and at 75 J/cm$^2$, respectively.

## Colony-forming-unit counting

After the final light exposure, the entire biofilm from each well was collected and placed into a 1-ml test tube, forming 200 µl of suspension. After meticulous vortexing (Vortex-Genie, Scientific Industries Inc, US), a serial dilution assay ranging from 1:1 to 1:100 000 was performed, using sterile ART filter tips (Thermo Scientific, Waltham, US). To enumerate the viable cells, 100 µl of resulting biofilm dilutions were then evenly spread over an entire BHI agar plates, using a sterile L-shape rod. According to the observed biofilm mass in the bottom of the well in each experiment, the dilutions were performed accordingly. As an example, treated biofilms were most usually serially diluted from 1:1 to 1:10$^4$, and controls usually from10$^5$ to 10$^6$, with single plating from each dilution. Typically, a dilution where CFU count on plate was between 30 to 800, was considered the most reliable and selected for analysis. The CFU 0 results were obtained with 1:1 dilution factor.

The plates were then assembled into the incubator, the bacteria were grown for 48 h, and the plates were photographed (Canon D5 DSLR camera with Canon EF 24–70 mm f/4L lens, Canon, Japan) on a light table (Artgraph Light Pad Revolution 80, Artograph Inc, US). The entire surface of each plate was included in the image. Colony-forming units were assessed with Image J software from the single photograph (National Institute of Health, US).

## Confocal scanning laser microscope (CSLM)

The structural organization of the biofilm was examined with confocal fluorescence imaging with a Leica TCS CARS SP 8X microscope (Leica Microsystems, Wetzlar, Germany), using

HC PL APO CS2 20X/0.75 numerical-aperture multi immersion and HX PL APO CS2 63X/ 1.2 numerical-aperture water immersion objectives. The imaged biofilms were stained using a live/dead BacLight bacterial viability kit (Molecular Probes. Invitrogen, Eugene, Oregon, USA). The stains were prepared according to manufacturer directions and were left to incubate at room temperature and in the dark for 15 min prior to examination under the confocal scanning laser microscope (CSLM). Light excitation was performed with a two-laser system, a 488 nm Argon laser, and a 561 nm DPSS laser, the emission windows configured to exclude the excitation wavelength of the two lasers and to meet the emission wavelength of the live/ dead fluorescence marker. The emission window for the 488-nm laser was set at 500 nm—530 nm, and for the 561-nm laser, at 620 nm—640 nm.

## Absorption spectroscopy assessment of ICG within a *Streptococcus mutans* pellet

One ml of *S. mutans* suspension with 0.46 OD, corresponding approximately to $100x10^6$ CFUs, was centrifuged for 5 minutes at 8000 rpm (Heraeus Megafuge 1.0, Thermo Scientific, Waltham, US) to the bottom of a 2-ml Eppendorf tube to form a 10-μl pellet. The supernatant was removed and replaced by a 1-mg/ml ICG solution to establish a total volume of 1 ml. The pellet was then mixed into the solution by vortexing for 60 seconds and left to incubate for 10 minutes, after which it was washed twice by centrifugation of the bacteria into a pellet and by replacing and vortexing the supernatant into a 0.9% NaCl solution. This was followed by re-centrifugation at 8000 rpm for 20 minutes. The *S. mutans*-formed pellet was vortexed into a fresh 0.9% NaCl solution to form 0.46 OD for an absorption spectroscopy analysis with a Gary 100 Bio UV-visible spectrophotometer (Varian Inc., Palo Alto, CA). For comparison purposes, an ICG 4 μg/ml NaCl 0.9% solution was used. The *S. mutans* 0.46 OD 0.9% NaCl solution suspension was used as a reference sample for the ICG/*S. mutans* suspension and 0.9% NaCl for the ICG 4 μg/ml NaCl 0.9% solution.

For antibacterial effectivity assessment, the 200ul of washed and resuspended ICG incubated bacteria solution was divided into 5 wells of Nunclon Delta well plates (Thermo Fisher Scientific Inc, US) followed by excitation with 810nm NIR LED light. Light intensity was 100 mW/cm^2 and the total delivered light dose was 100 J/cm^2. Control samples were prepared identically to treated samples excluding the ICG incubation step.

## Statistical analysis

Colony-forming-unit counts were compared with the non-parametric Mann-Whitney U-test, using GraphPad Prism 8 software (GraphPad Software, San Diego, US).

## Results

### Single treatment of one-day biofilm

The one-day *S. mutans* biofilm exhibited significantly reduced viability when exposed to aBL, from a median of $19x10^6$ CFUs (the range being $2.6x10^6$-$34x10^6$ CFUs) to a median of $1.65x10^6$ CFUs (the range being $0.08x10^6$- $10x10^6$ CFUs) (p = 0.0455, Mann Whitney U-test). Antibacterial photodynamic therapy administered with an 810-nm light together with ICG resulted in a markedly better efficiency than did aBL, with a three logarithmic scale reduction in alive bacteria counts compared to the control biofilm, showing a median of $1.3x10^3$ CFUs (the range being $0.3x10^3$-$47x10^3$ CFUs) (p = 0.0025, Mann-Whitney U-test). However, when aBL and aPDT were combined via the dual-light aPDT, the median of alive bacteria decreased to 0 CFU (the range being $0-0.7x10^3$ CFUs) (p = 0.0043, Mann-Whitney test). The bactericidal

effect of dual-light aPDT was significantly more efficient when compared to aPDT or aBL provided separately (p = 0.0064, p = 0.0022, respectively, Mann-Whitney U-test), see Fig 1.

## Single treatment of four-day biofilm

In the four-day biofilm model, where a single dose of aBL or aPDT was given at the end of the biofilm maturation period, the aBL reduced the alive bacteria counts to a median of $2.7 \times 10^6$ CFUs (the range being $1.4 \times 10^6$-$17 \times 10^6$ CFUs), from a median of $28 \times 10^6$ CFUs (the range being $8.4 \times 10^6$-$68 \times 10^6$ CFUs) of the control biofilm (p = 0.0245, Mann-Whitney U-test). Again, the aPDT application was significantly more effective than the aBL one, leaving only a median of $2 \times 10^3$ CFUs (the range being $0.1 \times 10^3$-$11 \times 10^3$ CFUs) (p = 0.0022, aBL vs. aPDT, Mann-Whitney test). Similarly to the one-day biofilm test, the simultaneous application of aBL and aPDT in the dual-light aPDT group showed an improved bactericidal effect, leaving a median of $1.9 \times 10^3$ CFUs (the range being $0.1$-$48 \times 10^3$ CFUs). There was no statistical difference between aPDT and dual-light aPDT (p = 0.7381, Mann-Whitney U-test) in the single-dose treatment of the four-day biofilm model. In general, the bacterial viability was better after the single-dose treatment of the four-day biofilm, when compared to the respectively treated one-day biofilms, see Fig 2.

## Daily treatment of four-day biofilm

A daily-dose repetitive application of aBL on the four-day biofilm model showed significantly improved bacterial viability, with a median of $18 \times 10^6$ CFUs (the range being $13 \times 10^6$-$20 \times 10^6$ CFUs), when compared to the equivalent dose, i.e., a single-dose application of the aBL to a four-day matured biofilm, as described above (p = 0.0087, Mann-Whitney U-test). Similarly, a daily-dose repetitive application of aPDT left a median of $13.5 \times 10^3$ CFUs (the range being

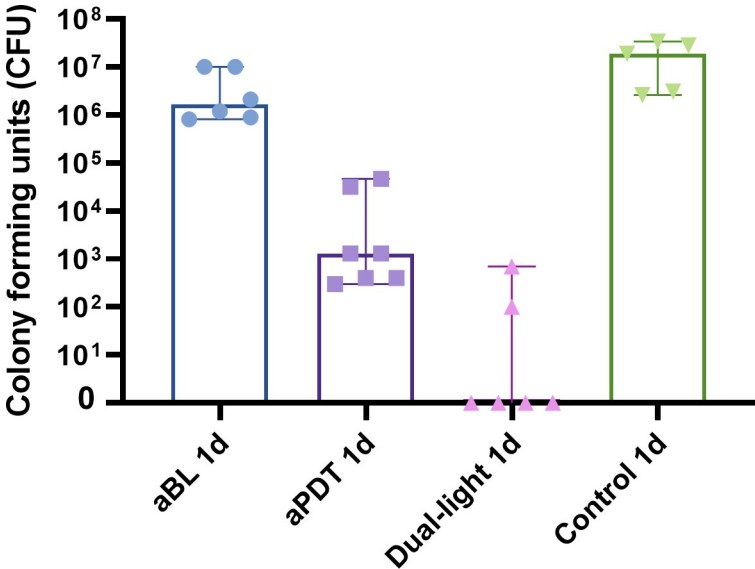

**Fig 1. Effect of single-dose application of aBL, aPDT or dual-light aPDT on one-day *S. mutans* biofilms.** A one-day *S. mutans* biofilm was treated with a single-dose application of aBL, aPDT, or dual-light aPDT. The total amount of light irradiance was the same at 100 mW/cm² for all three modalities (aBL vs. control, p = 0.045; aPDT vs. control, p = 0.0025; dual-light aPDT vs. control, p = p = 0.0043; aBL vs. dual-light aPDT, p = 0.0022; aPDT vs. dual-light aPDT, p = 0.0064; aBL vs. aPDT, p = 0.0012; Mann-Whitney U Test). The columns display medians. Each marking on the columns represents an independent assay. The T-bars show 95% confidence interval (CI). A detailed protocol and number of assays are shown at Table 1.

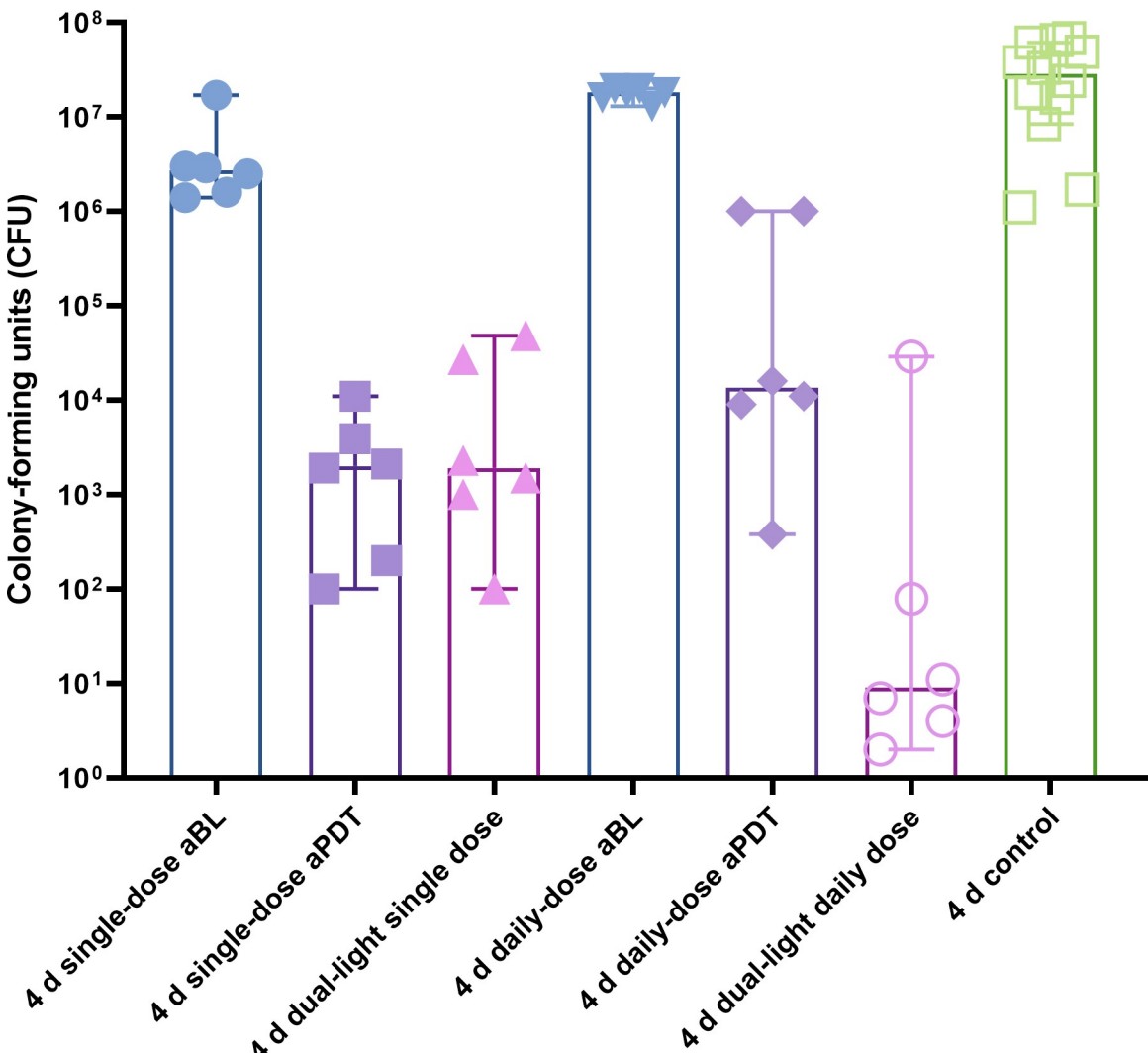

**Fig 2. Effect of single-dose or daily-dose application of aBL, aPDT or dual-light aPDT on four-day *S. mutans*.** A four-day biofilm was exposed to aBL, aPDT or dual-light aPDT as a single-dose exposure at the end of the biofilm maturation period or as a repetitive, daily-dose exposure repeating the same treatment dose. The columns display medians. Single-dose aBL vs. control, p = 0.025; single-dose aPDT vs. control, p = 0.0001; dual-light, single-dose aPDT vs. control, p = 0.0001; single- dose aBL vs. dual-light, single-dose aPDT, p = 0.0022; single-dose aPDT vs. dual-light, single-dose aPDT, p = 0.74; single-dose aBL vs. single-dose aPDT, p = 0.0022; daily-dose aBL vs. control, p = 0.35; daily-dose aPDT vs. control, p = 0.0001; dual-light, daily-dose aPDT vs. control, p = 0.0001; daily-dose aBL vs. dual-light, daily-dose aPDT, p = 0.00022; daily-dose aPDT vs. dual-light, daily-dose aPDT, p = 0.026; daily-dose aBL vs. daily-dose aPDT, p = 00022; single-dose aBL vs. daily-dose aBL, p = 0.0087; single-dose aPDT vs. daily-dose aPDT, p = 0.048; dual-light single dose vs. dual-light daily dose, p = 0.04; Mann-Whitney U Test.). The columns display medians. Each marking on the columns represents an independent assay. The T-bars show 95% confidence interval (CI). A detailed protocol and number of assays are shown at Table 1.

0.38x10³-1.0x10⁶ CFUs). This bacterial count is significantly more than in the four-day biofilm model, where the aPDT was applied as a single-dose treatment (p = 0.0476, Mann-Whitney U-test). However, the daily dose of combined aBL and aPDT in the dual-light aPDT group in the four-day biofilm model reduced the alive bacteria to a median of 9 CFUs (the range being 2-29x10³ CFUs). Thus, unlike the aBL or aPDT application, the repeated dual-light aPDT application significantly reduced the biofilm viability when compared to the equivalent dose, i.e., the single-dose application of the same treatment (p = 0.0411, Mann-Whitney U-test), see Fig 2.

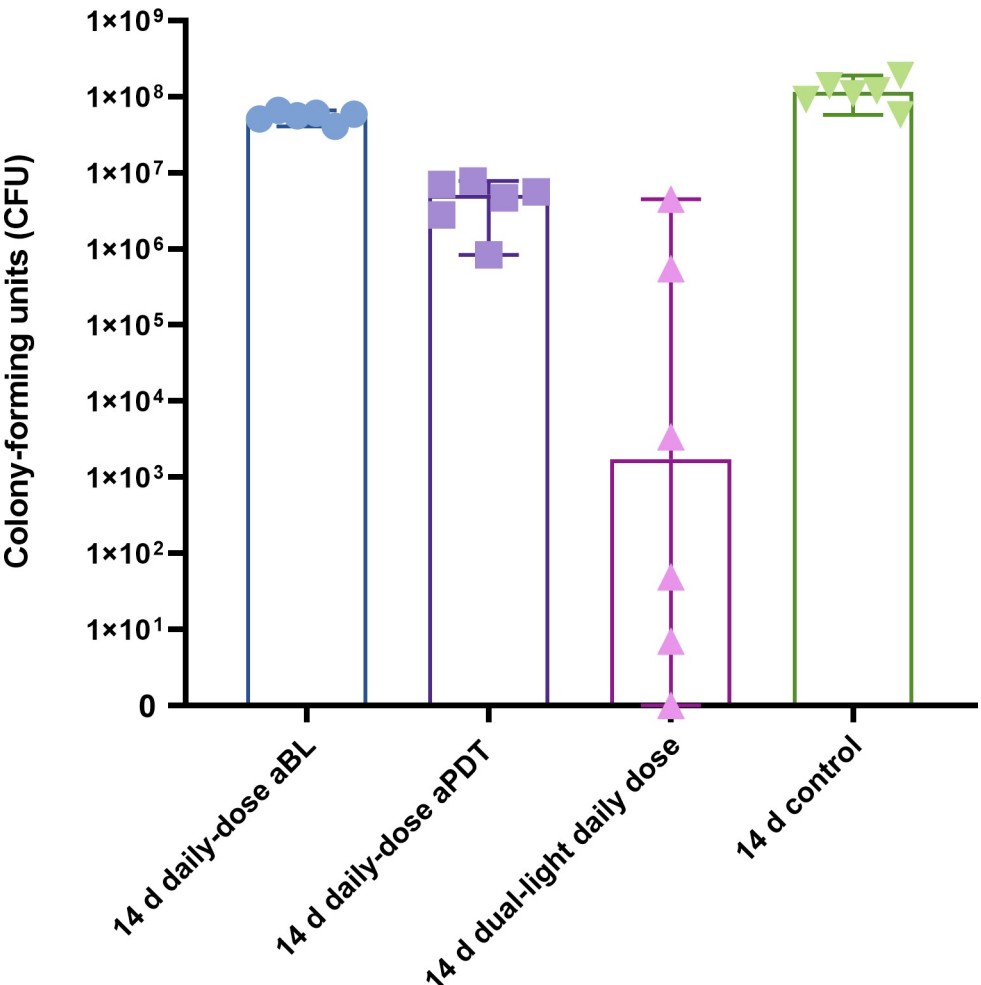

**Fig 3. Effect of daily-dose application of aBL, aPDT, or dual-light aPDT on fourteen-day *S. mutans* biofilm.** An extended daily-dose study protocol of fourteen days was established to test the ability of the biofilm to adapt to a repetitive aBL or aPDT application (aBL vs control, p = 0.02; aPDT vs. control, p = 0.0022; aPDT vs. aBL, p = 0.0022). The dual-light, daily-dose light application showed the most effective antibacterial effect (dual-light vs. aBL, p = 0.0022, dual-light vs. aPDT, p = 0.0087; dual-light vs. control, p = 0.0022, Mann-Whitney U-Test). The columns display medians. Each marking on the columns represents an independent assay. The T-bars show 95% confidence interval (CI). A detailed protocol and number of assays are shown at Table 1.

## Daily treatment of fourteen-day biofilm

A daily-dose repetitive application of aBL in the fourteen-day biofilm model showed, similarly to the four-day biofilm model, significantly improved bacterial viability, with a median of $57.5 \times 10^6$ CFUs (the range being $41 \times 10^6$-$66 \times 10^6$ CFUs), when compared to the four-day biofilm treated with single-dose aBL (p = 0.0022), or even the four-day daily-dose repetitive aBL application (p = 0.0022). However, the fourteen-day daily-dose of aBL The daily dose of aPDT in the fourteen-day biofilm model similarly improved viability of the biofilm, with a median of 5.1 CFUs (the range being $0.8$–$7.8 \times 10^6$ CFUs), when compared to the four-day biofilm treated by a single dose of aPDT (p = 0.0022) or a daily dose of aPDT (0.0087). In the fourteen-day biofilm model, again, the dual-light aPDT outperformed aBL or aPDT, with ongoing improvement in the bactericidal effect, leaving only 1725 CFUs (the range being $1$–$7.9 \times 10^6$ CFUs), see Fig 3. No significant difference was observed among the dual-light fourteen-day daily-dose

biofilm treatment, the four-day single-dose treatment, or the four-day daily-dose treatment (p = 0.8182 and p = 0.4199, respectively).

## Changing the energy ratio of aBL to aPDT lights while keeping the irradiance exposure constant

We analyzed the impact of changing the ratio of aBL to aPDT concerning dual-light aPDT antibacterial efficacy. In the one-day *Streptococcus mutans* biofilm, applying dual-light aPDT at a 1:1 irradiance ratio of aBL to aPDT light provided a median of 0 CFUs (the range being 0–500 CFUs), while a 3:1 irradiance ratio of aBL to aPDT light provided a median of 450 CFU count (the range being $0–7.8x10^3$ CFUs), and the 1:3 irradiance ratio of aBL to aPDT light provided a median of 700 CFUs (the range being $100-25x10^3$ CFUs). In the four-day biofilm model, having a single dose of dual-light aPDT treatment, the 1:1 irradiance ratio of aBL to aPDT light left a median of 100 CFUs (the range being $0–77 \ 10^3$ CFUs); the 3:1 irradiance ratio of aBL to aPDT light provided $10.3x10^3$ CFUs (the range being $0-780x10^3$ CFUs), and the 1:3 irradiance ratio of aBL to aPDT provided a median of 0 CFUs (the range being $0–2.2x10^3$ CFUs). Finally, In the daily, repetitive dual-light aPDT application, the 1:1 irradiance ratio of aBL to aPDT light left a median of 0 CFUs (the range being 0–400 CFUs); the 3:1 irradiance ratio of aBL to aPDT light left a median of $2.2x10^3$ CFUs (the range being $0-900x10^3$ CFUs); and the 1:3 light irradiance ratio of aBL to aPDT light left a median of 100 CFUs (the range being $0-740x10^3$ CFUs), see Fig 4.

## Confocal imaging of biofilms

With CSLM scanning, we were able to visualize the viability of the biofilm after each treatment protocol, see Fig 5. Multidimensional imaging of live (green) and dead (red) bacteria showed

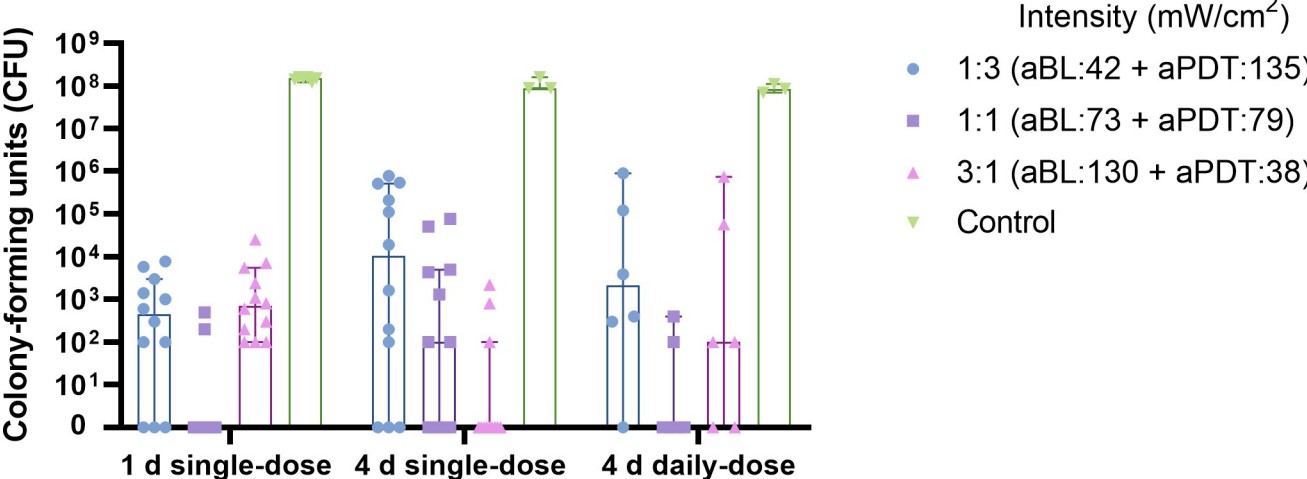

**Fig 4. Effect of change in the radiant exposure ratio of aBL to aPDT light in the antibacterial efficacy of the dual-light aPDT.** To assess the amount of blue light needed to increase the antibacterial efficacy of the aPDT, we established an experiment where the radiant exposure ratio of between aBL to aPDT was varied. Firstly, one-fourth of the total light irradiance was given as aBL (405 nm light) and three-fourths were given as aPDT (810 nm light). The exact amounts were 42 mW/cm² for the aBL and 135 mW/cm² for the aPDT, corresponding to a 1:3 ratio. Secondly, half of the total light irradiance was given as aBL (405 nm light) and half as aPDT (810 nm light), and the exact amounts were at 73 mW/cm² for the aBL and 79 mW/cm² for the aPDT, corresponding to 1:1 ratio. Thirdly, three-fourths of the total light irradiance were given as aBL (405nm light), and one fourth was given as aPDT (810nm light). The exact amounts in this case were at 130 mW/cm² for the aBL and 38 mW/cm² for the aPDT, corresponding to a 3:1 ratio. Single-day, single-dose, dual-light aPDT: 1:3 vs. 1:1, p = 0.003; 1:3 vs. 3:1, p = 0.43; 1:1 vs. 3:1; p<0.0001. Four-day, single-dose, dual-light aPDT: 1:3 vs. 1:1, p = 0.12; 1:3 vs. 3:1, p = 0.0057; 1:1 vs. 3:1; p = 0.067. Four-day, daily-dose aPDT: 1:3 vs. 1:1, p = 0.045; 1:3 vs. 3:1, p = 0.36; 1:1 vs. 3:1; p = 0.27; Mann-Whitney U Test.). The columns display medians. Each marking on the columns represents an independent assay. The T-bars show 95% confidence interval (CI). A detailed protocol and number of assays are shown at Table 1.

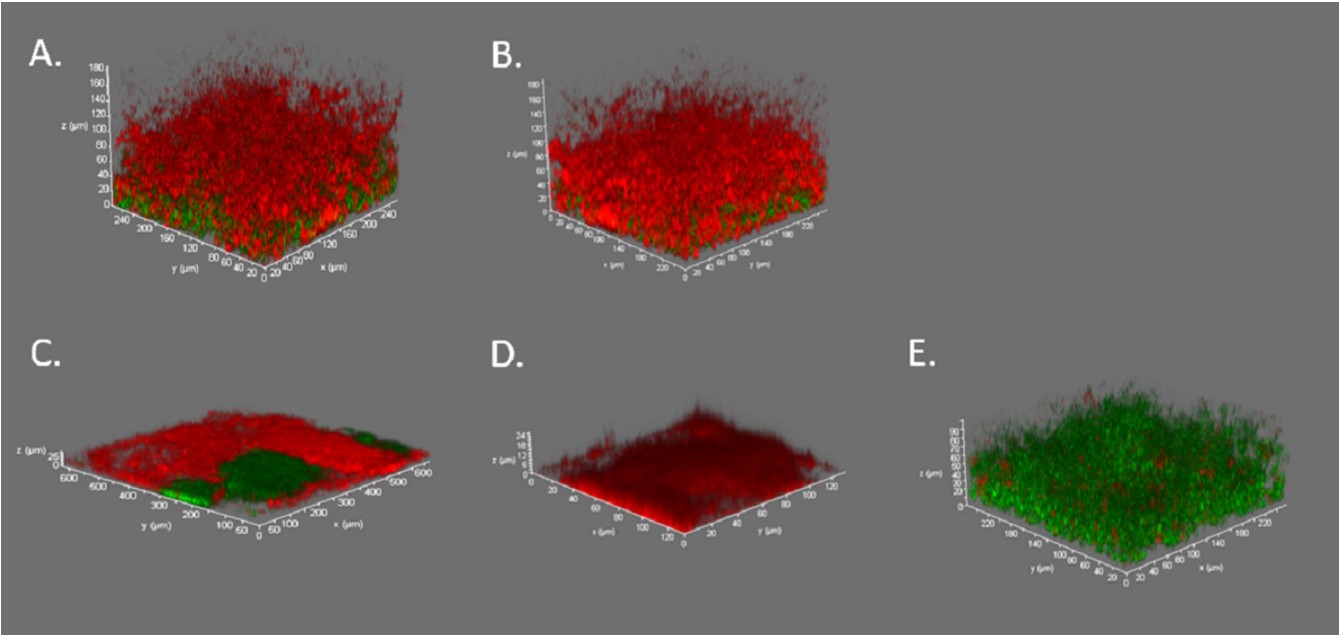

**Fig 5. Confocal 3D images of the four-day maturated biofilms stained with live/dead bacterial staining.** A. A single-dose application of aPDT. B. A single dose application of dual-light aPDT, C. A daily-dose application of aPDT for four days. D. A daily-dose application of dual-light aPDT for four days. E. A control four-day *S. mutans* biofilm.

most of the live bacteria located at the basal layer of the biofilm, suggesting that this area is more protected against any treatment. In the four-day biofilm model, the surviving bacteria were located at the bottom of a fully developed, thick biofilm. However, when the treatment was applied repetitively for four days, the biofilm appeared visually thinner and denser. The biofilms exposed to repetitive aPDT showed sporadic patchy areas where living bacteria were scattered. These patches could not be found in the dual-light aPDT-treated biofilms, see Fig 5.

### Absorption spectroscopy analysis of ICG adherence to *Streptococcus mutans*

Bacterial absorption difference between the ICG incubated *S. mutans* cells and the control *S. mutans* suspension in the 0.9% NaCl showed a 20-nm redshift of the absorption spectrum, when compared to the ICG absorption peak in water. Moreover, the absorption peak was lower, see Fig 6. The antibacterial effectivity of bacteria bound indocyanine green was investigated by exciting ICG bound bacteria with NIR light. Applying NIR light to ICG bound *S. mutans* bacterial significantly reduced bacterial viability, with a median of 21 CFUs (the range being 0–27 CFUs), when compared to the control sample with median of $13\times10^6$ (the range being $17\times10^6$-$9.7\times10^6$) (p = 0.0357, Mann-Whitney U test)., see Fig 7.

### Discussion

This is the first study demonstrating the superior efficacy of dual-light aPDT against *S. mutans* biofilm, when compared to aPDT or aBL. The simultaneous, synchronized application of an ICG/810 nm aPDT and 405 nm aBL resulted in a significantly improved antibacterial efficacy, the absolute CFU-count reduction constituting six logarithmic scales, and a persistent antibacterial effect. This persistent antibacterial effect has been nominated as substantivity, when applied to oral hygiene. Such substantivity, was not seen in the four-day repetitious aBL exposure, where bacterial CFU counts increased up to about five-fold when compared to a single-

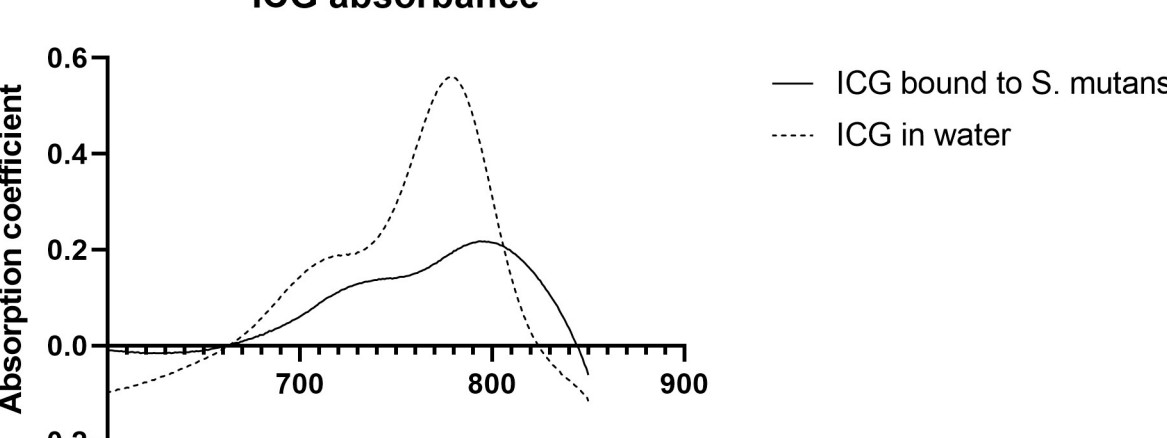

**Fig 6. Absorption spectrometry of ICG adherence to *S. mutans*.** The absorption spectrum of ICG bound to *S. mutans* and the absorption spectrum of free ICG dissolved in 0.9% NaCl.

dose aBL application. The ability of *S. mutans* to adapt to the repetitive aBL treatment eventually resulted in viability comparable to the control biofilm.

Similarly, the retained antibacterial action abated when repeated aPDT was applied. Although aPDT showed a significantly better antibacterial effect compared to aBL in *S. mutans* biofilm, the biofilm adapted to repeated exposure with increased CFU counts of up to 100-fold when compared to the single-dose aPDT treatment. The response to the repeated adverse environmental stimuli developed in the very early stage, within the first few days or doses of repeated exposure. The dual-light aPDT thus markedly outperformed both aPDT and aBL in efficacy, but most importantly, the synchronized use was able to suppress the ability of the biofilm to adapt to the external stress. This suppression provided the persistent action required if the method were to be adapted for clinical use in dentistry.

We assessed the relative amount of blue light needed to improve the efficacy of aPDT. The increase of aBL-part in the given radiant exposure of dual-light aPDT decreased the absolute amount of the aPDT effect because the total amount of radiant exposure was kept constant. This increase in the aBL was effective against older biofilm, showing the 3:1 aBL ratio as being the most effective against four-day-old biofilm, but the repetitive dual-light dosing was most effective when the aBL ratio stood at 1:1 with the 810-nm light. Of the different blue light spectrums, we chose to use 405 nm aBL for two main reasons. Firstly, the antibacterial efficacy of 405 nm light has been shown to outperform longer aBL wavelengths in several studies [21]. Secondly, even in the visible light spectrum, there are variations in harmfulness to eyes with different light wavelengths, and eye safety improves at 405 nm light, as compared to 450 nm light or other aBL alternatives.

Various photosensitizers and exciting light combinations have been used against dental biofilms [6]. Indocyanine green, widely used and tested in dentistry, has been approved by the Food and Drug Administration in the United States of America for this purpose. It has been shown as a rather weak singlet oxygen provider, but it does possess a temperature-raising antibacterial ability within a biofilm. After light absorption, the ICG molecule can reach the ground state by releasing the energy through three different pathways. Firstly, the energy can convert into a fluorescence emission ranging from 750 nm to 950 nm. The spectrum maximums are approximately 780 nm in water and 810 nm in blood. Secondly, part of the energy is transferred to an ICG triplet state via intersystem crossing, being able to produce reactive

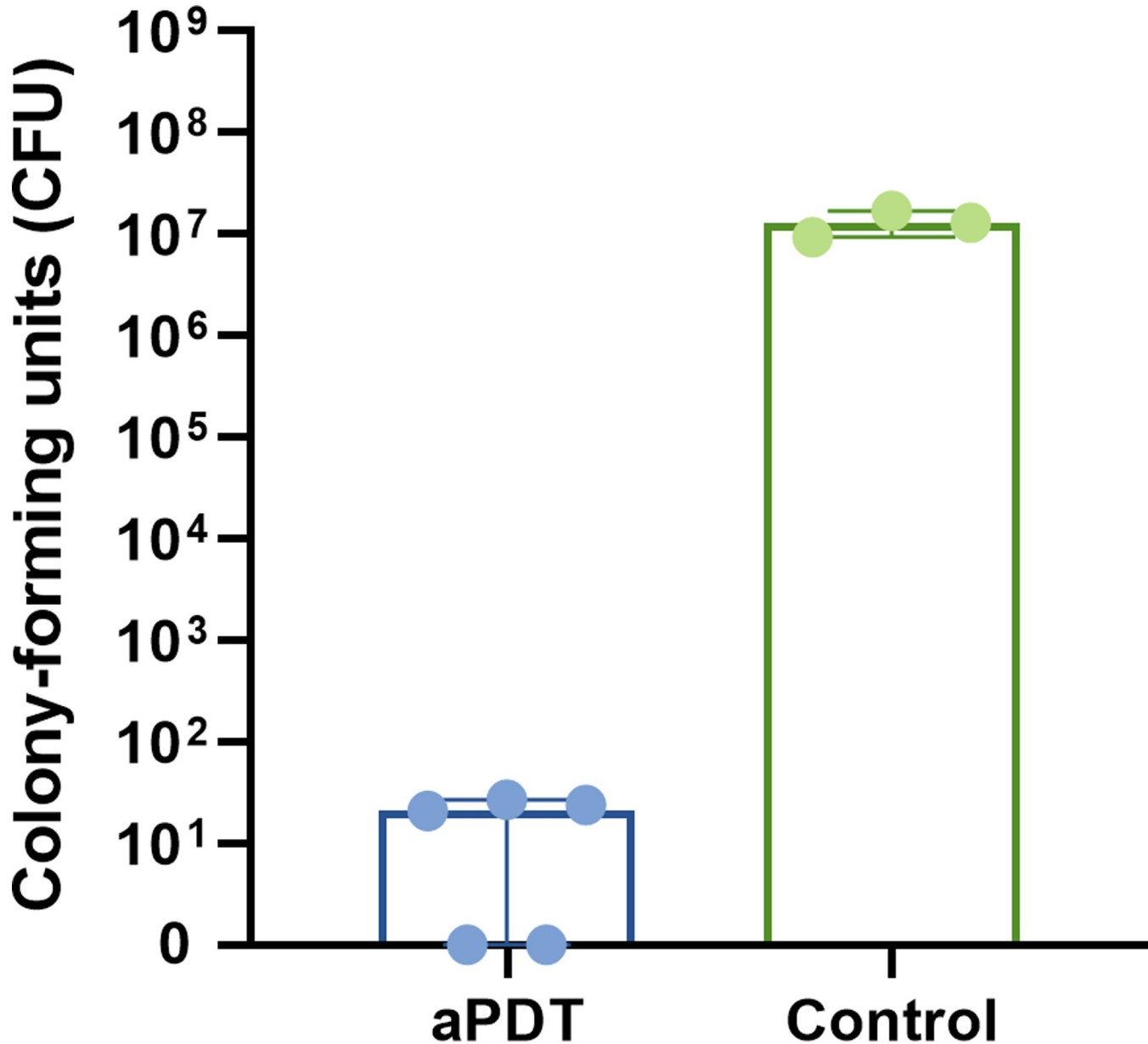

**Fig 7. Antibacterial activity of *S. mutans* bound Indocyanine green.** ICG dyed *S. mutans* was excited with 810 nm light and compared to non-ICG dyed control. The light exposure resulted in significant reduction of CFU levels (aPDT vs control, p = 0.0357, Mann-Whitney U test)

oxygen species. The yield of triplet formation of ICG is 14% in water, and 11% in an aqueous albumin solution. The quantum yield of triplet formation of ICG is sufficiently high for generating efficient reactive oxygen species, particularly singlet oxygen. Thirdly, the energy can be transformed into heat within the ICG molecule itself by internal conversion. It has been estimated that as much as 85% of the absorbed energy could be converted into heat [22]. The ability of ICG to produce antibacterial action through different mechanisms provides an attractive safety feature, especially if aPDT were to be administered frequently. The thermal antibacterial mechanisms could also provide an additional benefit when antibacterial efficacy is provided in deep periodontal pockets, where oxygen is not readily available.

We ruled out the macroscopic heating effect of the sample by confirming temperature levels below 35˚C degrees during each treatment. However, this confirmation does not rule out the temperature changes in the microenvironment, due to the inherent abilities of light absorbing ICG. Prokaryotic cells contain the same heat shock proteins as eukaryotic cells, enabling bacteria to cope against hostile environments. Heat stress has been shown to cause a distinct response in the *S. mutans* expression profile of multiple regulators and other functional genes [23,24]. The extracellular matrix architectural structure and the cells' ability to bind are impaired due to heat. Glycosyltransferase (gtf)-c, which is responsible for generating only partially water-soluble glucan, is upregulated; but gtf-b, which is responsible for producing the water-soluble external environment, is not. This change is detectable as early as five to ten minutes after the heat exposure. Similar early responses in the *S. mutans* expression profile can be seen in other heat-responsive genes, such as grpE, dnaK, and fruR. Eventually, the upregulation of clpE and clpP aid the survival of the cells in the harsh conditions, including increased oxidative stress [24]. Antibacterial blue light has also been shown to alter the gene expression of *S. mutans*, upregulating several genes such as gtfB, brp, smu630, and comDE. It has been shown to increase the susceptibility of bacteria to ROS [25], which can further explain not only the adaptive mechanisms but also the additive bactericidal effect of the simultaneous use of aBL and aPDT.

Indocyanine green has photodecomposition properties [26], which has been suggested to have effect on cell viability. The green color of the biofilm disappeared during each the light exposure, indicating ICG degradation. We found no improvement in the antibacterial action when the possible degradation product remnants were left in the wells after washing the illuminated IGC away. The longer the cells were incubated in the residual ICG or its photodecomposition products (up to 14 days), the more viable the biofilm was. Indocyanine green has also shown a low stability of in aqueous solutions, for which we changed a new ICG medium for each illumination to provide a fresh ICG substate for the aPDT action.

We used spectroscopy to measure the ICG absorption properties in the *S. mutans* solution. To our knowledge, no previous work has been published to investigate this. Indocyanine green has been previously shown to undergo redshift of the absorption maximum from 780 nm in water to 805 nm in plasma upon binding to albumin in blood plasma or when binding to lipid structures [27,28]. We found that a similar redshift was also present in planktonic *S. mutans* solution, providing evidence of ICG adhering to bacterial proteins and membrane structures. The lower intensity peak was due to the higher ICG concentration in water compared to that of *S. mutans*-bound ICG. The antibacterial effect of the adhered ICG was estimated by subsequent light excitation, with an 810 nm LED light source resulting in total bacteria-killing with light intensities from or above 20 J/cm$^2$. These results prove that the antibacterial activity is caused by the bacterial-bound ICG and not by the water-solubilized ICG. The antibacterial action of ICG was preserved in the bound ICG despite the lower ICG concentration, indicating the role of ICG binding to bacteria as the key treatment-targeting mechanism.

## Conclusions

The ability of *S. mutans* to cope with ICG/810 nm aPDT was effectively eliminated by adding simultaneous aBL to the treatment. Antibacterial blue light was markedly less bactericidal than aPDT when similar light doses were compared, but aBL improved the antibacterial effect of aPDT and provided a sustained effect in repeated antibacterial treatment. The mechanism is an issue yet to be resolved. Based on our findings the amount of added 405 nm aBL for an increased antibacterial efficacy should be at least 50% of the radiant exposure given. Along the aging of the biofilm, an increase in the relative amount of aBL would be beneficial. To sum up,

results from the present experiments open up new avenues for hypothesis generation and, more practically, for developing devices for biofilm control, especially in preventive dentistry.

## Acknowledgments

We would like to thank our laboratory technician Saija Perovuo for excellent work.

## Author Contributions

**Conceptualization:** Sakari Nikinmaa, Martti Vaara, Juha Rantala, Jukka Meurman, Tommi Pätilä.

**Data curation:** Sakari Nikinmaa.

**Formal analysis:** Sakari Nikinmaa, Tommi Pätilä.

**Funding acquisition:** Sakari Nikinmaa.

**Investigation:** Sakari Nikinmaa, Tommi Pätilä.

**Methodology:** Petri Auvinen, Juha Rantala, Timo Sorsa, Jukka Meurman, Tommi Pätilä.

**Project administration:** Sakari Nikinmaa, Tommi Pätilä.

**Resources:** Sakari Nikinmaa.

**Supervision:** Heikki Alapulli, Petri Auvinen, Martti Vaara, Timo Sorsa, Jukka Meurman, Tommi Pätilä.

**Validation:** Sakari Nikinmaa, Martti Vaara, Jukka Meurman.

**Visualization:** Sakari Nikinmaa, Tommi Pätilä.

**Writing – original draft:** Sakari Nikinmaa, Juha Rantala, Timo Sorsa, Jukka Meurman, Tommi Pätilä.

**Writing – review & editing:** Sakari Nikinmaa, Heikki Alapulli, Petri Auvinen, Martti Vaara, Juha Rantala, Esko Kankuri, Timo Sorsa, Jukka Meurman, Tommi Pätilä.

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
