## [Decision Letter · Decision Letter 0]

4 Feb 2020

PONE-D-20-01567

Dual-light photodynamic therapy administered daily provides a sustained antibacterial effect on biofilm and prevents Streptococcus mutans adaptation

PLOS ONE

Dear Mr Patila,

Thank you for submitting your manuscript to PLOS ONE. After careful consideration, we feel that it has merit but does not fully meet PLOS ONE’s publication criteria as it currently stands. Therefore, we invite you to submit a revised version of the manuscript that addresses the points raised during the review process.

Please provide number of replicates and error bars, remove results description from figure legends, address photodecomposition of ICG and other comments of Rev 2.

We would appreciate receiving your revised manuscript by Mar 20 2020 11:59PM. To enhance the reproducibility of your results, we recommend that if applicable you deposit your laboratory protocols in protocols.io, where a protocol can be assigned its own identifier (DOI) such that it can be cited independently in the future. For instructions see: http://journals.plos.org/plosone/s/submission-guidelines#loc-laboratory-protocols

We look forward to receiving your revised manuscript.

Kind regards,

Michael R Hamblin

Academic Editor

PLOS ONE

Journal Requirements:

2. Thank you for stating the following in the Competing Interests section: "No"

We note that one or more of the authors are employed by a commercial company: Koite Health Oy

3. Thank you for stating the following in your Competing Interests section:  "No"

4. Thank you for stating the following financial disclosure: "No"

a)    Please provide an amended Funding Statement that declares *all* the funding or sources of support received during this specific study (whether external or internal to your organization) as detailed online in our guide for authors at http://journals.plos.org/plosone/s/submit-now. 

b)    Please state what role the funders took in the study.  If any authors received a salary from any of your funders, please state which authors and which funder. If the funders had no role, please state: "The funders had no role in study design, data collection and analysis, decision to publish, or preparation of the manuscript."

6. Please upload a copy of Figure 7, to which you refer in your text on page 15. If the figure is no longer to be included as part of the submission please remove all reference to it within the text.

Reviewers' comments:

Reviewer's Responses to Questions

**Comments to the Author**

1. Is the manuscript technically sound, and do the data support the conclusions?

Reviewer #1: Yes

Reviewer #2: Yes

2. Has the statistical analysis been performed appropriately and rigorously? 

Reviewer #1: Yes

Reviewer #2: Yes

3. Have the authors made all data underlying the findings in their manuscript fully available?

Reviewer #1: Yes

Reviewer #2: Yes

4. Is the manuscript presented in an intelligible fashion and written in standard English?

Reviewer #1: Yes

Reviewer #2: Yes

5. Review Comments to the Author

Reviewer #1: Manuscript No: PONE-D-20-01567

Manuscript Title: Dual-light photodynamic therapy administered daily provides a sustained antibacterial effect on biofilm and prevents Streptococcus mutans adaptation.

In this study, the authors tested the efficacy of repeated treatments of single and combined aPDT and aBL in an S. mutans biofilm-model.

The manuscript is on an emergent topic, photodynamic therapy as an alternative approach to inactivate pathogenic bacteria, addressing one of the major difficulties in developing effective protocols against bacterial biofilms.

The manuscript is well structured along the different sections. The results are credible, and the discussion is clear and focused. However, the experimental plan is not adequately described. There is no indication about the number of independent assays performed for each situation. Also, the number of replicates of the detection technique is not indicated. Moreover, no standard deviation values are showed in the figures.

The legends of figures are very long, the description of the results should be removed. For instance, in the Figure 1 legend, the text “Dual-light aPDT was more effective than peers, when one-day biofilm was treated” and “The aBL reduced the number of living bacteria significantly, but the antibacterial effect of aPDT was markedly better than that of aBL. The combination of the two, the dual-light aPDT, provided significantly greater antibacterial activity“, should be deleted.

The title can be something like: “Effect of single-dose application of aBL, aPDT or dual-light aPDT on S. mutans biofilms” …..

The standard deviation must be added to the figure and the number of independent assays must be indicated in the legend.

Minor editing on English is necessary.

Reviewer #2: In this manuscript the combination of aPDT with and without blue light illumination was investigated. The manuscript is of interest, but contains some drawbacks, which must be revised in order to improve the manuscript.

critics:

line 46-ff: "...Biofilms were scraped, diluted......After reincubation, ......, and confocal 3D biofilm immaging was performed". This must be specified in more details. The aPDT treated biofilm was scraped, and recultured again to Biofilms. What is the aim of this experiment? This is unclear and must be revised for a better understanding.

line 52: "milder response": How is this defined? Unclear. Please insert an adequate definition.

Materials and Methods section: I miss information about the number of independent experiments. How many values per each parameter did you investigate? Furthermore, I recommend to insert a table showin all investigated conditions and parameters. Which combination experiments were done using aPDT +/- blue light +/- ICG. Such information should be insereted within the revised version. I think it is a must have to understand how often the biofilms were illluminated.

Figure 1: Single illumination ???? Not clear. What is the difference between Dual-light 1D illumination vs. 4d dual-light daily dose (Figure 2). In both cases you achieved >5log10 steps of CFU reductions. This must be explained in more details.

Figure 4: The values within the brackets??? applied light intensities??? Please revise.

Please insert a more detailed summary. Which combination would you now recommend based of the given results? A clear take home message must be inserted.

General: It is known, that ICG is damaged during illumination. Did you measure absorption spectra prior and after illumination of ICG? For more details please see: Light-induced decomposition of indocyanine green. Engel E, Schraml R, Maisch T, Kobuch K, König B, Szeimies RM, Hillenkamp J, Bäumler W, Vasold R. Invest Ophthalmol Vis Sci. 2008 May;49(5):1777-83. doi: 10.1167/iovs.07-0911.

Do you think, that such a decomposition is influencing the efficacy of bacteria killing? Decomposition of ICG is a crucial factor, isn't it?

6. PLOS authors have the option to publish the peer review history of their article (what does this mean?). If published, this will include your full peer review and any attached files.

Reviewer #1: Yes: Adelaide Almeida

Reviewer #2: No

---

## [Author Response · Author response to Decision Letter 0]

17 Mar 2020

PLOS ONE 

Comments for editor and the reviewers:

Revision of the manuscript: ‘Dual-light photodynamic therapy administered daily provides a sustained antibacterial effect on biofilm and prevents Streptococcus mutans adaptation’

We kindly thank the Editor and the reviewers for their time and efforts to revise our manuscript. We found the advice and thoughts very relevant and are very glad for the help provided.

Editor

Q1: Please provide number of replicates and error bars, remove results description from figure legends, address photodecomposition of ICG and other comments of Rev 2.

A1: We have provided the number of replicates and error bars, and removed results description from figure legends, addressed photodecomposition of ICG and the other comments of Rev 2.

Q2: A: We have rephrased our financial disclosure in the cover letter. 

Q3: To enhance the reproducibility of your results, we recommend that if applicable you deposit your laboratory protocols in protocols.io.

A3: We deposited our laboratory protocol at protocols.io, with the DOI: dx.doi.org/10.17504/protocols.io.bdp2i5qe

Q4& A4: Please find our rebuttal letter including responds to each point raised by the editor and reviewers. This letter has been uploaded as separate file and labelled 'Response to Reviewers'. We have uploaded a marked-up copy of your manuscript that highlighting the changes made to the original version. This file has been uploaded as separate file and labelled 'Revised Manuscript’ with ‘Track Changes'. The unmarked version of our revised paper without tracked changes has been uploaded as separate file and labelled 'Manuscript'

Editorial additional requirements:

Q1. We have rechecked, that our manuscript meets PLOS ONE's style requirements

Q2-4. We have provided the Competing Interest Statement and the Financial Disclosure in the Cover letter. 

Q5. We note that you have included the phrase “data not shown” in your manuscript. Unfortunately, this does not meet our data sharing requirements. PLOS does not permit references to inaccessible data. We require that authors provide all relevant data within the paper, Supporting Information files, or in an acceptable, public repository. Please add a citation to support this phrase or upload the data that corresponds with these findings to a stable repository (such as Figshare or Dryad) and provide and URLs, DOIs, or accession numbers that may be used to access these data. Or, if the data are not a core part of the research being presented in your study, we ask that you remove the phrase that refers to these data.

A5: We have provided all relevant data within the paper, the “data not shown”, regarding the ICG adherence to mutans bacteria, is represented as figure 7, and discussed in text: 

Line 355: “The antibacterial effectivity of bacteria bound indocyanine green was investigated by exciting ICG bound bacteria with NIR light. Applying NIR light to ICG bound S. mutans bacterial significantly reduced bacterial viability, with a median of 21 CFUs (the range being 0-27 CFUs), when compared to the control sample with median of 13x106 (the range being 17x106-9.7x106) (p=0.0357, Mann-Whitney U test)., see Fig 7.”

And line 364: “Fig 7. Antibacterial activity of S. mutans bound Indocyanine green. ICG dyed Streptococcus mutans was excited with 810nm light and compared to non-ICG dyed control. The light exposure resulted in significant reduction of CFU levels (aPDT vs control, p = 0.0357, Mann-Whitney U test)”

Q6. Please upload a copy of Figure 7, to which you refer in your text on page 15. If the figure is no longer to be included as part of the submission, please remove all reference to it within the text.

A6: The Fig 7 was not part of the original submission and this reference has been removed from the text. However, we added information according to Answer 5 and a new Fig 7 file is referred in the text as described. 

Reviewer 1

1.The manuscript is well structured along the different sections. The results are credible, and the discussion is clear and focused. However, the experimental plan is not adequately described. There is no indication about the number of independent assays performed for each situation. Also, the number of replicates of the detection technique is not indicated. Moreover, no standard deviation values are showed in the figures.

We had a short mention of the number of independent assays in the manuscript, line 109: “A minimum of six biofilms for each experiment were grown”. We had also provided this data in our figures as each dot represent an independent assay. Obviously, this was not enough, and we have additionally provided a ‘table 1’ to show the number of independent assays and included the study protocol for each experiment for more clear understanding for the reader.

Added in the text, line 120: “Study protocols are presented in Table 1.”

Table 1 legend: “Study protocols”

2. Also, the number of replicates of the detection technique is not indicated. 

This is a good question.

1. All independent experiments were plated separately. 

Added in the text, line 118: “In each setting, the last treatment was followed by plating each well onto separate brain heart infusion (BHI)-agar dishes for colony-forming unit (CFU) counting”

2. Because of the experience of our laboratory technician with over 20 years of bacterial laboratory work in GLP level laboratories, we could adjust the CFU assessment according to each sample, without the need for full dilution series. We have performed these studies for three years, which has provided us a good understanding of the procedure.

Added in the text, line 180: “According to the observed biofilm mass in the bottom of the well in each experiment, the dilutions were performed accordingly. As an example, treated biofilms were most usually serially diluted to from 1:1 to 1:104 and controls usually 105-106, with a single plating from each dilution.”

3. Each plate was entirely photographed and the CFU count was performed using J-image by a single analysis. 

We revised the text for more clarity, line 176: “After the final light exposure, the entire biofilm from each well was collected and placed into a 1-ml test tube, forming 200 μl of suspension. After meticulous vortexing (Vortex-Genie, Scientific Industries Inc, US), a serial dilution assay ranging from 1:1 to 1:100 000 was performed, using sterile ART filter tips (Thermo Scientific, Waltham, US). To enumerate the viable cells, 100 μl of resulting biofilm dilutions were then evenly spread over an entire BHI agar plates, using a sterile L-shape rod. According to the observed biofilm mass in the bottom of the well in each experiment, the dilutions were performed accordingly. As an example, treated biofilms were most usually serially diluted from 1:1 to 1:104, and controls usually from105 to 106. Typically, a dilution where CFU count on plate was between 30 to 800 was considered the most reliable and selected for analysis.”

3. Moreover, no standard deviation values are showed in the figures.

Our data does not present Gaussian distribution. We have provided the statistical comparisons with non-linear tests, including median and range of each dataset. Thus, we are statistically unable to provide standard deviation of mean. We added 95% Confidence Interval (CI) error bars in the figures, while the colums represent the medians, and the dots show the range.

5.The legends of figures are very long, the description of the results should be removed. For instance, in the Figure 1 legend, the text “Dual-light aPDT was more effective than peers, when one-day biofilm was treated” and “The aBL reduced the number of living bacteria significantly, but the antibacterial effect of aPDT was markedly better than that of aBL. The combination of the two, the dual-light aPDT, provided significantly greater antibacterial activity“, should be deleted. The title can be something like: “Effect of single-dose application of aBL, aPDT or dual-light aPDT on S. mutans biofilms”

A5: This is a good comment. We have removed thoroughly the results description in all figure legends. 

a. Fig 1 legend. Change in the text, line 242: The figure title has been changed according to suggestion. The descriptive text has been removed. 

b. Fig 2 legend: Change in the text, line 261: The figure title has been changed to “Effect of single-dose or daily-dose application of aBL, aPDT or dual-light aPDT on four-day S. mutans” The descriptive text has been removed. 

c. Fig 3 legend: Change in the text, line 298: The figure title has been changed to “Effect of daily-dose application of aBL, aPDT or dual-light aPDT on fourteen-day S. mutans biofilm” The descriptive text has been removed. 

Fig 4 legend: Change in the text, line 321: The figure title has been changed to: Effect of change in the relative irradiance between aBL and aPDT light in the antibacterial efficacy of the dual-light aPDT. To assess the amount of blue light needed to increase the antibacterial efficacy of the aPDT, we established an experiment where the relative irradiance ratios between the aPDT and aBL were varied. Firstly, one-fourth of the total light irradiance was given as aBL (405nm light) and three-fourths were given as aPDT (810nm light). The exact amounts were 42mW/cm2 for the aBL and 135 mW/cm2 for the aPDT, corresponding to a 1:3 ratio. Secondly, half of the total light irradiance was given as aBL (405nm light) and half as aPDT (810nm light), and the exact amounts were at 73 mW/cm2 for the aBL and 79 mW/cm2 for the aPDT, corresponding to 1:1 ratio. Thirdly, three-fourths of the total light irradiance were given as aBL (405nm light), and one fourth was given as aPDT (810nm light). The exact amounts in this case were at 130mW/cm2 for the aBL and 38 mW/cm2 for the aPDT, corresponding to a 3:1 ratio. Single-day, single-dose, dual-light aPDT: 1:3 vs. 1:1, p=0.003; 1:3 vs. 3:1, p=0.43; 1:1 vs. 3:1; p<0.0001. Four-day, single-dose, dual-light aPDT: 1:3 vs. 1:1, p=0.12; 1:3 vs. 3:1, p=0.0057; 1:1 vs. 3:1; p=0.067. Four-day, daily-dose aPDT: 1:3 vs. 1:1, p=0.045; 1:3 vs. 3:1, p=0.36; 1:1 vs. 3:1; p=0.27; Mann-Whitney U Test. ). The columns display medians. Each marking on the columns represents an independent assay. The T-bars show 95% confidence interval (CI). A detailed protocol and number of assays are shown at Table 1.

d. ” The descriptive text of results has been removed.

e. Fig 5: the title was changed:, line 346: “Confocal 3D images of the four-day maturated biofilms stained with live/dead bacterial staining”. The descriptive text of results has been removed.

f. Fig 6: The title has changed, line 361; “Absorption spectrometry of ICG adherence to S. mutans”

Q6. The standard deviation must be added to the figure and the number of independent assays must be indicated in the legend.

A6: The number of independent assays are now provided in the table 1. The error bars as 95% confidence interval have been provided in the figures. 

8.Minor editing on English is necessary.

We have re-edited the language

Reviewer #2:

Q1: In this manuscript the combination of aPDT with and without blue light illumination was investigated. The manuscript is of interest, but contains some drawbacks, which must be revised in order to improve the manuscript.

A1: We greatly appreciate your comments and efforts to improve our manuscript. 

Q2: line 46-ff: "...Biofilms were scraped, diluted......After reincubation, ......, and confocal 3D biofilm immaging was performed". This must be specified in more details. The aPDT treated biofilm was scraped, and recultured again to Biofilms. What is the aim of this experiment? This is unclear and must be revised for a better understanding.

A2: Thank you for the comment. We have thoroughly revised the abstract text. 

A2: line 52: "milder response": How is this defined? Unclear. Please insert an adequate definition.

A3: Good comment. We revised the abstract text according to suggestion. 

Q4:Materials and Methods section: I miss information about the number of independent experiments. How many values per each parameter did you investigate? Furthermore, I recommend to insert a table showin all investigated conditions and parameters. Which combination experiments were done using aPDT +/- blue light +/- ICG. Such information should be insereted within the revised version. I think it is a must have to understand how often the biofilms were illluminated

A4: We have included the number of the independent experiments in the figures, but also provided a table for this (Table1.), which is needed to guide the reader through the article. 

Q4: Figure 1: Single illumination ???? Not clear. What is the difference between Dual-light 1D illumination vs. 4d dual-light daily dose (Figure 2). In both cases you achieved >5log10 steps of CFU reductions. This must be explained in more details.

A4: We have provided a table to clarify this question, where we conclude the actual number of illuminations and the amount of light in each illumination. This simplifies the actual finding in our paper: If we give a single illumination on a biofilm, most of the bacteria are dead. But if we repeat this treatment with the same intensity for several days, the biofilm gets better.

Q5: Figure 4: The values within the brackets??? applied light intensities??? Please revise.

A5: We have provided this information in the figure legend. line 322: “To assess the amount of blue light needed to increase the antibacterial efficacy of the aPDT, we established an experiment where the relative irradiance ratios between the aPDT and aBL were varied. Firstly, one-fourth of the total light irradiance was given as aBL (405nm light) and three-fourths were given as aPDT (810nm light). The exact amounts were 42mW/cm2 for the aBL and 135 mW/cm2 for the aPDT, corresponding to a 1:3 ratio. Secondly, half of the total light irradiance was given as aBL (405nm light) and half as aPDT (810nm light), and the exact amounts were at 73 mW/cm2 for the aBL and 79 mW/cm2 for the aPDT, corresponding to 1:1 ratio. Thirdly, three-fourths of the total light irradiance were given as aBL (405nm light), and one fourth was given as aPDT (810nm light). The exact amounts in this case were at 130mW/cm2 for the aBL and 38 mW/cm2 for the aPDT, corresponding to a 3:1 ratio.” The descriptive text of results has been removed. Additionally, the Table 1 is provided to clarify the message.

Please insert a more detailed summary. Which combination would you now recommend based of the given results? A clear take home message must be inserted.

A5: We revised the conclusion, line 453: “The ability of S. mutans to cope with ICG/810-nm aPDT was effectively eliminated by adding simultaneous aBL to the treatment. Antibacterial blue light was markedly less bactericidal than aPDT when similar light doses were compared, but aBL improved the antibacterial effect of aPDT and provided a sustained effect in repeated antibacterial treatment. The mechanism is an issue yet to be resolved. The amount of added 405nm aBL for an increased antibacterial efficacy would be at least 50% of the total light energy given. Along the aging of the biofilm, an increase in the relative amount of aBL would be beneficial. To sum up, results from the present experiments open up new avenues for hypothesis generation and, more practically, for developing devices for biofilm control, especially in preventive dentistry.”

Q6: General: It is known, that ICG is damaged during illumination. Did you measure absorption spectra prior and after illumination of ICG? For more details please see: Light-induced decomposition of indocyanine green. Engel E, Schraml R, Maisch T, Kobuch K, König B, Szeimies RM, Hillenkamp J, Bäumler W, Vasold R. Invest Ophthalmol Vis Sci. 2008 May;49(5):1777-83. doi: 10.1167/iovs.07-0911.Do you think, that such a decomposition is influencing the efficacy of bacteria killing? Decomposition of ICG is a crucial factor, isn't it?

A6: This is an insightful comment, which we appreciate. We are well aware of the photo-decomposition of ICG and their properties. We also acknowledge the low stability of ICG in aqueous solutions. According to our study, we see no improvement in the antibacterial action when the possible degradation product remnants were left in the wells after washing the illuminated IGC away. For each illumination we changed a new ICG medium for the described ICG incubation period to provide a fresh ICG substate for the aPDT action. The green color of the biofilm disappeared during the light exposure, indicating ICG degradation after each exposure. If the photodecomposition would have an effect, the bacterial biofilm would have suffered the most in cases where the biofilm was in longest contact with the byproducts, but we saw the opposite. 

1. We added in the discussion a paragraph about this: Line 431” Indocyanine green has photodecomposition properties [25], which has been suggested to have effect on cell viability. The green color of the biofilm disappeared during each the light exposure, indicating ICG degradation. We found no improvement in the antibacterial action when the possible degradation product remnants were left in the wells after washing the illuminated IGC away. The longer the cells were incubated in the residual ICG or its photodecomposition products (up to 14 days), the more viable the biofilm was. Indocyanine green has also shown a low stability of in aqueous solutions, for which we changed a new ICG medium for each illumination to provide a fresh ICG substate for the aPDT action.”

---

## [Decision Letter · Decision Letter 1]

12 Apr 2020

PONE-D-20-01567R1

Dual-light photodynamic therapy administered daily provides a sustained antibacterial effect on biofilm and prevents Streptococcus mutans adaptation

PLOS ONE

Dear Mr Patila,

Thank you for submitting your manuscript to PLOS ONE. After careful consideration, we feel that it has merit but does not fully meet PLOS ONE’s publication criteria as it currently stands. Therefore, we invite you to submit a revised version of the manuscript that addresses the points raised during the review process.

We would appreciate receiving your revised manuscript by May 27 2020 11:59PM. To enhance the reproducibility of your results, we recommend that if applicable you deposit your laboratory protocols in protocols.io, where a protocol can be assigned its own identifier (DOI) such that it can be cited independently in the future. For instructions see: http://journals.plos.org/plosone/s/submission-guidelines#loc-laboratory-protocols

We look forward to receiving your revised manuscript.

Kind regards,

Michael R Hamblin

Academic Editor

PLOS ONE

Reviewers' comments:

Reviewer's Responses to Questions

**Comments to the Author**

1. If the authors have adequately addressed your comments raised in a previous round of review and you feel that this manuscript is now acceptable for publication, you may indicate that here to bypass the “Comments to the Author” section, enter your conflict of interest statement in the “Confidential to Editor” section, and submit your "Accept" recommendation.

Reviewer #2: All comments have been addressed

Reviewer #3: (No Response)

2. Is the manuscript technically sound, and do the data support the conclusions?

Reviewer #2: Yes

Reviewer #3: Yes

3. Has the statistical analysis been performed appropriately and rigorously? 

Reviewer #2: Yes

Reviewer #3: Yes

4. Have the authors made all data underlying the findings in their manuscript fully available?

Reviewer #2: Yes

Reviewer #3: Yes

5. Is the manuscript presented in an intelligible fashion and written in standard English?

Reviewer #2: Yes

Reviewer #3: Yes

6. Review Comments to the Author

Reviewer #2: The revised version is now improved. All points and suggestions of the reviewer are fulfilled. The revised version can be now recommended for publication.

Reviewer #3: An interesting manuscript looking at the potentiated antimicrobial effects using a combination of PDT and aBL against S. mutans biofilm.

Before acceptance of the manuscript, the following points should be addressed.

Line 97 – the citations only seems to include resistance to PDT and not aBL, should add some aBL citations.

Line 106 –were the replicates performed independently on 6 different days?

Line 140 –it would be nice to have the absorption spectrum of the photosensitizer

Line 152 – add emission spectra of all light sources used

Line 154 – should be written 100 mW/cm2 instead of 100mW/cm2 – this should be changed for all within the manuscript.

Line 164 – what is meant by ‘relative amounts?’ the authors need to be very clear

Line 173 -what was the purpose of adjusting the irradiances using these ratios? was it  to ensure similar durations of exposure? Was duration of the treatment most important to the authors? I would have thought distributing ratio’s in terms of radiant exposure may have made more sense.

Line 239 - 0 colonies on the plate does not mean 0 CFU, you are limited by the threshold for detection based on the dilution factor. Please amend your figures to factor in the limit of detection.

Line 308 – again please be more specific than ‘relative amounts’.

7. PLOS authors have the option to publish the peer review history of their article (what does this mean?). If published, this will include your full peer review and any attached files.

Reviewer #2: No

Reviewer #3: No

---

## [Author Response · Author response to Decision Letter 1]

18 Apr 2020

We kindly thank the editor, Mr Hamblin, and the reviewers for their time and efforts to revise our manuscript. We found the advice and thoughts very relevant and are very glad for the help provided.

Reviewer #2

1. The revised version is now improved. All points and suggestions of the reviewer are fulfilled. The revised version can be now recommended for publication.

We kindly thank you for your time and efforts to improve our publication.

Reviewer #3:

An interesting manuscript looking at the potentiated antimicrobial effects using a combination of PDT and aBL against S. mutans biofilm. Before acceptance of the manuscript, the following points should be addressed.

We kindly thank you for your interest towards our work and we greatly appreciate your help in improving our paper. The clear presentation of these comments helped us greatly to address the issues.

1. Line 97 – the citations only seems to include resistance to PDT and not aBL, should add some aBL citations.

A: We added the following citations, line 97: 

Guffey SJ, Payne W, Jones T, and Martin K. Evidence of Resistance Development by Staphylococcus aureus to an In Vitro, Multiple Stage Application of 405 nm Light from a Supraluminous Diode Array Photomedicine and Laser Surgery. 2013, Vol. 31, No. 4

Yucheng Wang, Ying Wang,a,Yuguang Wang,a,e Clinton K. Murray,f, Michael R. Hamblin, David C. Hooper, and Tianhong Dai. Antimicrobial blue light inactivation of pathogenic microbes: state of the art Drug Resist Updat. 2017 Nov; 33-35: 1–22.

2. Line 106 –were the replicates performed independently on 6 different days?

We have performed all the experiments of a given study protocol during the same day, simultaneously. This enabled us to use the same S. mutans suspension, identical light exposure parameters and provided exactly similar laboratory conditions. We believe that the samples to be comparable with each other, the conditions should be kept as similar as possible.

Added in the text line 109: All the replicants in each study protocol were performed during the same day, simultaneously. This enabled to use the same S. mutans suspension, identical light exposure parameters, and provided exactly similar laboratory conditions.

3. Line 140 –it would be nice to have the absorption spectrum of the photosensitizer

A: Absorption spectrum of ICG can be seen under the title: ‘Absorption spectroscopy analysis of ICG adherence to Streptococcus mutans’ and seen in Fig 6. 

Added in the text, line 142: Absorption spectrum of ICG is provided below.

4. Line 152 – add emission spectra of all light sources used

A: We provided Table 2 to provide this information. 

Added in the text, line 153: The emission spectra of the used light sources are presented in Table 2.

5. Line 154 – should be written 100 mW/cm2 instead of 100mW/cm2 – this should be changed for all within the manuscript.

A: This change has been made. We did the same change from 100J/cm2 to 100 J/cm2 as well.

6. Line 164 – what is meant by ‘relative amounts?’ the authors need to be very clear

A: Good point, thank you. This definitely clarifies the message. We went through the paper to clear this problem.

Added in the text, line 166, among other changes: "We also tested the antibacterial efficacy of dual-light treatment in terms of different radiant exposure ratios of 405 nm to 810 nm, when the total amount of light was kept constant at 100 J/cm2."

7. Line 173 -what was the purpose of adjusting the irradiances using these ratios? was it to ensure similar durations of exposure? Was duration of the treatment most important to the authors? I would have thought distributing ratio’s in terms of radiant exposure may have made more sense.

A: This is a good point. The amount of radiant exposure (J/m2) was the constant of interest, and the changing irradiance (W/m2) ratios of aBL and aPDT light sources reflect to that. The text just doesn’t follow this idea clearly enough. We have carefully revised the manuscript, clearing the idea of radiant exposure being the changing ratio.

Changes in the text have been revised throughout, see especially paragraph starting at line 169.

8. Line 239 - 0 colonies on the plate does not mean 0 CFU, you are limited by the threshold for detection based on the dilution factor. Please amend your figures to factor in the limit of detection.

According to the observed biofilm mass in the bottom of the well in each experiment, the dilutions were performed accordingly. As an example, treated biofilms were most usually serially diluted to from 1:1 to 1:10^4 and controls usually 10^5-10^6, with a single plating from each dilution.

The CFU 0 results were obtained with 1:1 dilution factor. Biofilm from well was mechanically removed to 200ul of buffer from which 100ul was plated to agar. If bacteria were viable, they should form a countable colony to agar in two days. This means that in these l zero results there were no viable bacteria in plated 100ul of the sample. All experiments were done with identical biofilm extraction procedure and in 6 repeats to mitigate small statistical errors in sampling. The sampling errors are small compared to logarithmic differences between different treatments. 

Added in the text, line 190: The CFU 0 results were obtained with 1:1 dilution factor.

9. Line 308 – again please be more specific than ‘relative amounts’.

A: Thank you for these comments, helping us to communicate our results more clearly. 

We have thoroughly revised the manuscript regarding ‘relative amounts’, irradiance and radiant exposure. Please see the Manuscript track changes-version, line 53, line 171, line 315, line 331, line 398 and line 470.

---

## [Editor Report · Decision Letter 2]

22 Apr 2020

Dual-light photodynamic therapy administered daily provides a sustained antibacterial effect on biofilm and prevents Streptococcus mutans adaptation

PONE-D-20-01567R2

Dear Dr. Patila,

We are pleased to inform you that your manuscript has been judged scientifically suitable for publication and will be formally accepted for publication once it complies with all outstanding technical requirements.

With kind regards,

Michael R Hamblin

Academic Editor

PLOS ONE
---

## [Editor Report · Acceptance letter]

23 Apr 2020

PONE-D-20-01567R2 

Dual-light photodynamic therapy administered daily provides a sustained antibacterial effect on biofilm and prevents *Streptococcus mutans* adaptation 

Dear Dr. Pätilä:

I am pleased to inform you that your manuscript has been deemed suitable for publication in PLOS ONE. Congratulations! Your manuscript is now with our production department. 

With kind regards,

on behalf of

Dr. Michael R Hamblin 

Academic Editor

PLOS ONE